# Age-related response to mite parasitization and viral infection in the honey bee suggests a trade-off between growth and immunity

**Virginia Zanni**[ORCID]*, **Davide Frizzera, Fabio Marroni**[ORCID]**, Elisa Seffin, Desiderato Annoscia, Francesco Nazzi**\*

Dipartimento di Scienze AgroAlimentari, Ambientali e Animali (DI4A), Università degli Studi di Udine, Udine, Italy

\* virginia.zanni@uniud.it (VZ); francesco.nazzi@uniud.it (FN)

## Abstract

Host age at parasites' exposure is often neglected in studies on host-parasite interactions despite the important implications for epidemiology. Here we compared the impact of the parasitic mite *Varroa destructor*, and the associated pathogenic virus DWV on different life stages of their host, the western honey bee *Apis mellifera*. The pre-imaginal stages of the honey bee proved to be more susceptible to mite parasitization and viral infection than adults. The higher viral load in mite-infested bees and DWV genotype do not appear to be the drivers of the observed difference which, instead, seems to be related to the immune-competence of the host. These results support the existence of a trade-off between immunity and growth, making the pupa, which is involved in the highly energy-demanding process of metamorphosis, more susceptible to parasites and pathogens. This may have important implications for the evolution of the parasite's virulence and in turn for honey bee health. Our results highlight the important role of host's age and life stage at exposure in epidemiological modelling. Furthermore, our study could unravel new aspects of the complex honey bee-*Varroa* relationship to be addressed for a sustainable management of this parasite.

**Data Availability Statement:** All relevant data are within the paper and its Supporting Information and raw data files.

## 1. Introduction

Parasitism is a common lifestyle in the biosphere. Roughly 50% of the organisms live at the expense of others [1] negatively affecting their fitness and survival [2].

The damage that a parasite causes to its host (i.e. parasite's virulence) is shaped by evolution [3] and affected by several ecological factors such as temperature [4], shortage of nutrients [5], host density [6] as well as by host and parasite's genetics [7]. The host's age or the life stage at the exposure can also influence the parasite's virulence [8], nevertheless, with the exception of human studies, this element is often neglected in the investigations on host-parasite interactions [9, 10]. As a matter of fact, models not considering age are often utilized for epidemiological studies and considered representative of the whole population [9]. However, the host population is often characterized by individuals of different age classes living in close contact with each other (e.g. social insects) and several parasites are able to parasitize hosts at different

**Funding:** This research was funded by the European Union's Horizon 2020 research and innovation program, under Grant Agreement No. 773921 (PoshBee) and by the Italian Ministry of University, PRIN 2017 - UNICO (2017954WNT). The funders had no role in study design, data collection and analysis, decision to publish, or preparation of the manuscript.

**Competing interests:** The authors have declared that no competing interests exist.

ages with different virulence [9]. In the case of invertebrates, age-structured epidemiological models are practically inexistent [8] and only a few studies delved into these aspects. Izhar and Ben-Ami [9] investigated how the host age at the exposure affects the virulence of the parasite *Pasteuria ramose* on the crustacean *Daphnia magnae*. It appeared that exposure at different host ages, significantly affects the parasite transmission and its overall virulence, highlighting how the optimal virulence can change with host age. According to the higher susceptibility showed by the younger *Daphnia*, Izhar and Ben-Ami suggested that the parasite may have a higher reproductive potential in younger hosts since this stage is characterized by a longer life expectation, an immature immune system and lower costs of clearance. In fact, immune system activation is costly for the host [11] and the investment in growth may subtract resources from the immunity causing higher susceptibility in younger individuals [12]. According to current interpretations [9], this could force a trade-off between immune function and development. In the particular case of social insects, the suggested trade-off between growth and immunity may be compensated by the adoption of a "social immune system" resulting from the cooperation between group members aimed at limiting the risk of parasites' transmission that arises from group living [13].

Honey bees can be affected by a number of parasites such as viruses, bacteria, fungi, trypanosomes and mites. Disregarding the distinction between sexes and castes and focusing only on worker bees, the honey bee colony is characterized by a mixture of life stages and age classes with distinct roles and susceptibility to parasites. For example, the bacteria *Penibacillus larvae* and *Melissococcus pluton*, causing the American and European foulbrood, the fungus *Ascosphaera apis*, causing Chalkbrood, and the sacbrood virus (SBS), causing the disease named accordingly, only affect the brood while the tracheal mite *Acarapis woodi* and the microsporidian *Nosema apis* and *ceranae*, causing acariosis and nosemiasis respectively, only affect the adults. Among the parasites affecting bees at all stages, a pre-eminent role is played by the ectoparasitic mite *Varroa destructor*. The mite life cycle is strictly synchronized with that of the honey bees [14] and consists of a reproductive phase at the expense of the pre-imaginal stages of bees and a phoretic phase at the expense of adults [14]. Mite infestation has several direct effects on honey bees, including water loss [15], decreased flight performance [16] and behavioral modifications [17, 18]; furthermore, through the feeding activity, the mite can transmit bacteria and viruses [19, 20]. Among the plethora of viruses vectored by the mite [21–23], the genotypes A and B of Deformed wing virus (DWV) are particularly detrimental for the colonies [24]. The mite and the DWV are linked in a mutualistic symbiosis and the mite, besides vectoring DWV, can promote its replication [25, 26]. DWV can also be vertically transmitted from queens to their eggs or from infected sperm to the zygote [27], and horizontally, among different bee developmental stages, through bee products as honey, pollen and fresh royal jelly [28, 29], contact with faeces [30], food exchange and cannibalism [31]. DWV can infect adults and all pre-imaginal stages of drones, queens and workers but viral levels differ between developmental phases and castes, with workers' pupae showing the highest viral level and adult drones the lowest [32]. Despite the symptoms (i.e. deformed wings, bloated abdomen, decreased body size) of overt infection being exclusive to the adult bees infected during the larval and pupal stages [33], the different virulence according to honey bee stage at the time of the exposure is still not defined.

To counteract the effects of the parasitization, the honey bee can rely on an innate immune system [34] which, however, is characterized by a reduced number of immunity genes if compared with solitary insects [35]. Albeit reduced, the pool of bees' immunity genes is efficiently involved in the antimicrobial and antiviral responses [35, 36]; furthermore, bees show an efficient RNA interference response that represents their broadest antiviral defence mechanism [37, 38].

A previous study highlighted how the bee's immune response towards an artificial infection with viable *Escherichia coli* bacteria changes according to the life stage, showing that larvae and adults are better equipped than pupae to react to infections [39]. The reason may lie in the previously described trade-off between growth and immunity. This could be particularly important for a holometabolous insect like *A. mellifera* which, during the metamorphosis from pupa to adult, needs to re-allocate the resources stored during the larval stage [40], a process which could well limit the available immune resources. Honey bee larvae, reared in open cells and tended several times a day by a number of nurses, are highly exposed to the risk of infections and are therefore endowed with an efficient immune response, complemented with various antimicrobial compounds supplied by larval food [41]. Instead, bee pupae, developing inside a sealed cell, preventing contact with nest mates and previously treated with an antibiotic substance like propolis [42], probably need to invest fewer resources in an active immune system saving them for the demanding last moult [43]. In conclusion, although the reduced investment in immune resources during the development is partly compensated by the social protection from infections (e.g. use of propolis, enclosure of the developing pupae into the protected brood cell), developing pupae represent a particularly susceptible life stage to parasites such as the *Varroa* mite that spends its reproductive phase in the sealed brood cell, feeding on the pupa [14], nullifying the protection offered by the operculated cell.

In this study we compare the impact of the main ectoparasite of the western honey bee, the mite *Varroa destructor*, on different life stages of its host, then we assess the immune response of those life stages and interpret the observed response against the current trade-off hypothesis. In view of the close relationship between the mite and the vectored DWV, we also consider this pathogenic virus within the same interpretative framework.

We conclude that the possible trade-off between growth and immunity makes the pre-imaginal stages of honey bees particularly susceptible to an important ecto-parasite and the symbiotic virus with critical consequences for honey bee survival and potential implications for the evolution of the parasite's virulence.

## 2. Results

### 2.1 Effects of mite infestation on honey bee survival according to the host's life stage

To assess the effect of *Varroa* infestation on honey bee survival, we exploited two complementary approaches. In the first one, we compared honey bees that were mite-infested or not, either during the pupal stage or at the adult stage. Then, we carried out another experiment whereby some bees were infested or not at the pupal stage, and the emerging bees from the uninfested group were either mite-infested or not at the adult stage.

**2.1.1 Effect of mite infestation during the pupal stage on the survival of bees.** *Varroa* mite infestation during the pupal stage significantly reduced the survival of the eclosing adult bees (Fig 1, Wilcoxon test: chi square = 6.4, *df* = 1, $p < 0.01$); the reduction in median longevity caused by mite infestation during the pupal stage was 53% (median survival: control bees = 19 days, mite-infested bees = 9 days).

**2.1.2 Effect of mite infestation at the adult stage on the survival of bees.** Mite infestation at the adult stage caused a significant reduction in bees' survival (Fig 2, Wilcoxon test: chi square = 7.82, *df* = 1, $p < 0.01$); the reduction in median longevity caused by mite infestation during the adult stage was 12% (median survival: control bees = 25 days; mite-infested bees = 22 days).

**2.1.3 Effect of the time of mite infestation on bees' survival.** Mite infestation affected negatively honey bees' survival, however the magnitude of the effect changed accordingly to

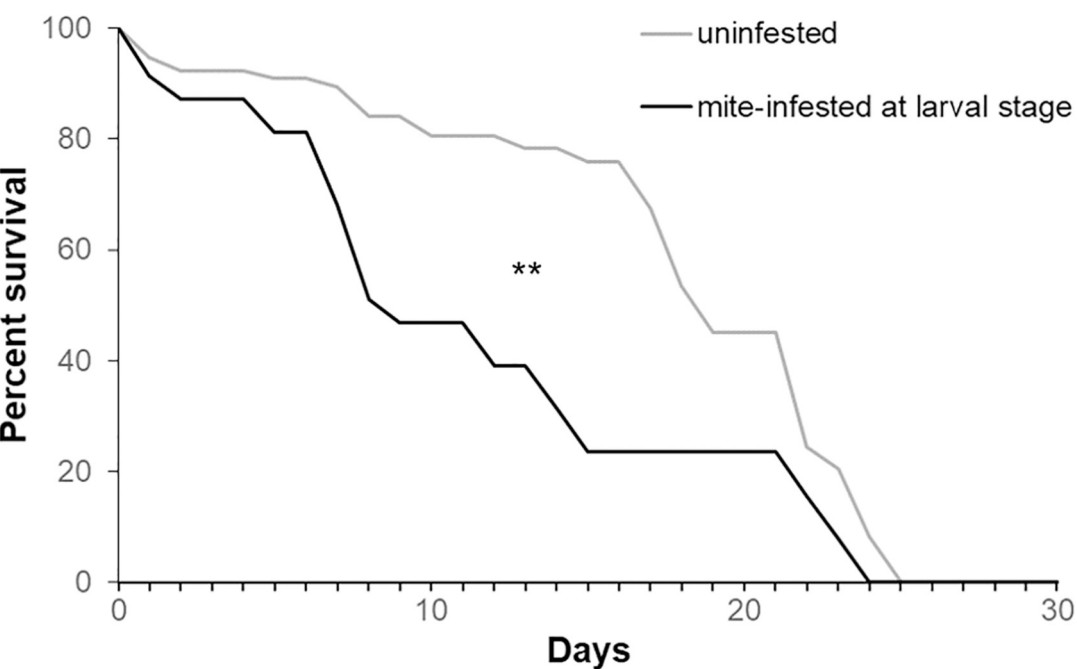

**Fig 1. Survival of honey bees mite-infested during the pupal stage.** Survival curves of the adult bees artificially infested or not with one *Varroa* mite at the L5 stage and maintained under lab conditions (Wilcoxon test. Chi square = 6.4; *df* = 1; *p* < 0.01, uninfested bees: n = 103, mite-infested bees: n = 76).

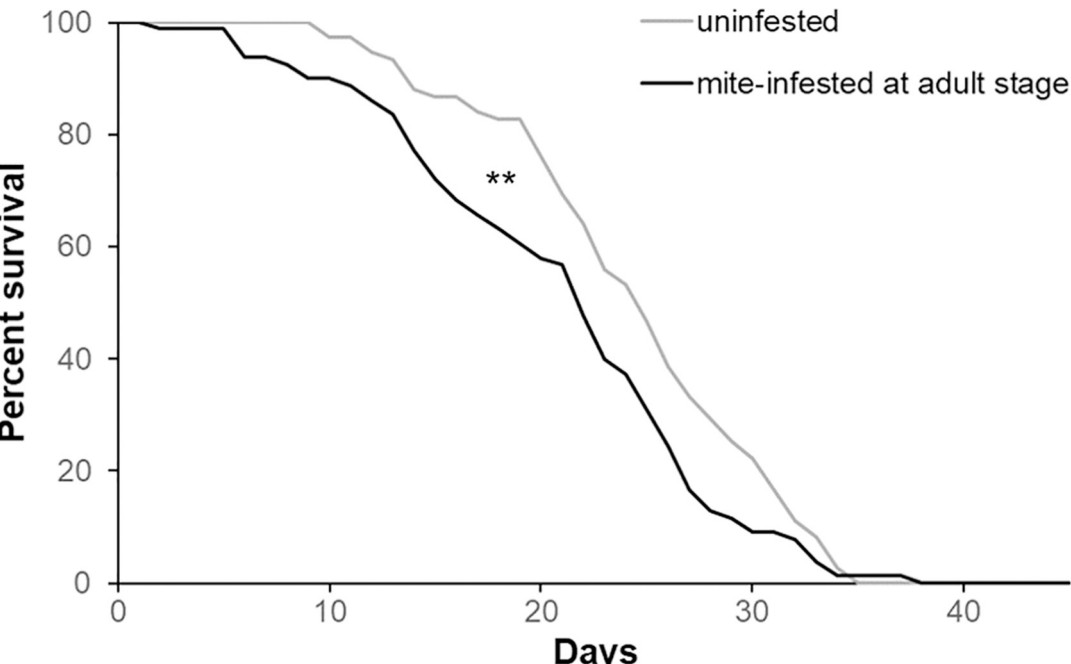

**Fig 2. Survival of the honey bees mite-infested at the adult stage.** Survival curves of the adult bees artificially infested or not with one *Varroa* mite at the emergence and maintained under lab conditions (Wilcoxon test. Chi square = 7.82; *df* = 1; *p* < 0.01, uninfested bees: n = 82, mite-infested bees: n = 83).

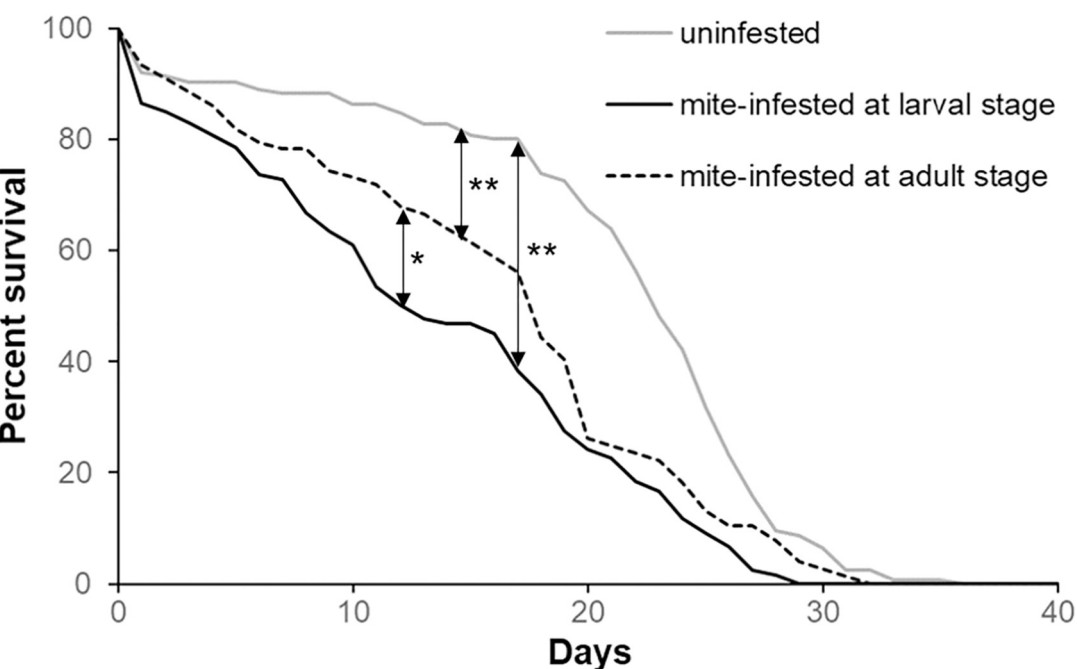

**Fig 3. Effects of the infestation's time on bees' survival.** Survival curves of the adult bees uninfested and artificially infested with one *Varroa* mite at the L5 stage or at the emergence and maintained under lab condition (Wilcoxon test. Infestation at larval stage vs control: chi square = 52.3, $df$ = 1, $p < 0.01$; infestation at adult stage vs control: chi square = 20.12, $df$ = 1, $p < 0.01$; infestation at larval stage vs infestation at adult stage: chi square = 3.70, $df$ = 1, $p = 0.05$, uninfested bees: n = 164, bees mite-infested at the larval stage: n = 140, bees mite-infested at the adult stage: n = 121).

the host stage. For this reason, the parasite effect was further investigated with an additional experiment whereby we directly compared the survival of adult bees collected from the same colony at the larval stage and mite-infested either during the pupal stage or immediately after the emergence. As observed before, honey bees' survival was reduced by mite parasitization in both cases (Fig 3, Wilcoxon test, infestation at larval stage: chi square = 52.3, $df$ = 1, $p < 0.01$; infestation at adult stage: chi square = 20.12; $df$ = 1; $p < 0.01$); however, bees infested during the pupal stage survived less than bees infested at the adult stage (Fig 3, Wilcoxon test: chi square = 3.70, $df$ = 1, $p = 0.05$) confirming the higher impact of mite infestation during the pupal stage observed above.

**2.1.4 Mite's effects on DWV load in parasitized bees.** DWV level at the emergence was significantly higher in bees that were mite-infested during the pupal stage as compared to uninfested bees (Fig 4, Mann-Whitney U test. $U = 22$, $df = 1$, $p = 0.02$).

In the case of mite infestation at the adult stage, there was no significant difference in DWV level in bees infested with *Varroa* and the control (Mann-Whitney U test. $U = 27$, $df = 1$, $p = 0.3$) likely because of the high variability observed in the mite-infested group.

RNAseq analysis performed on the adult bees collected at the larval stage and mite-infested either during the pupal stage or immediately after the emergence highlighted a higher proportion of reads mapping on the DWV genome in mite-infested individuals (Fig 5, Mann-Whitney U test. Bees infested when larvae vs control: $U = 0$, $df = 1$, $p < 0.01$; bees infested when adults vs control: $U = 0$, $df = 1$, $p < 0.01$). The proportion of DWV mapping reads in bees infested at the larval stage was not significantly different from that of bees infested at the adult stage (Fig 5, Mann-Whitney U test. $U = 13$, $df = 1$, $p = 0.07$).

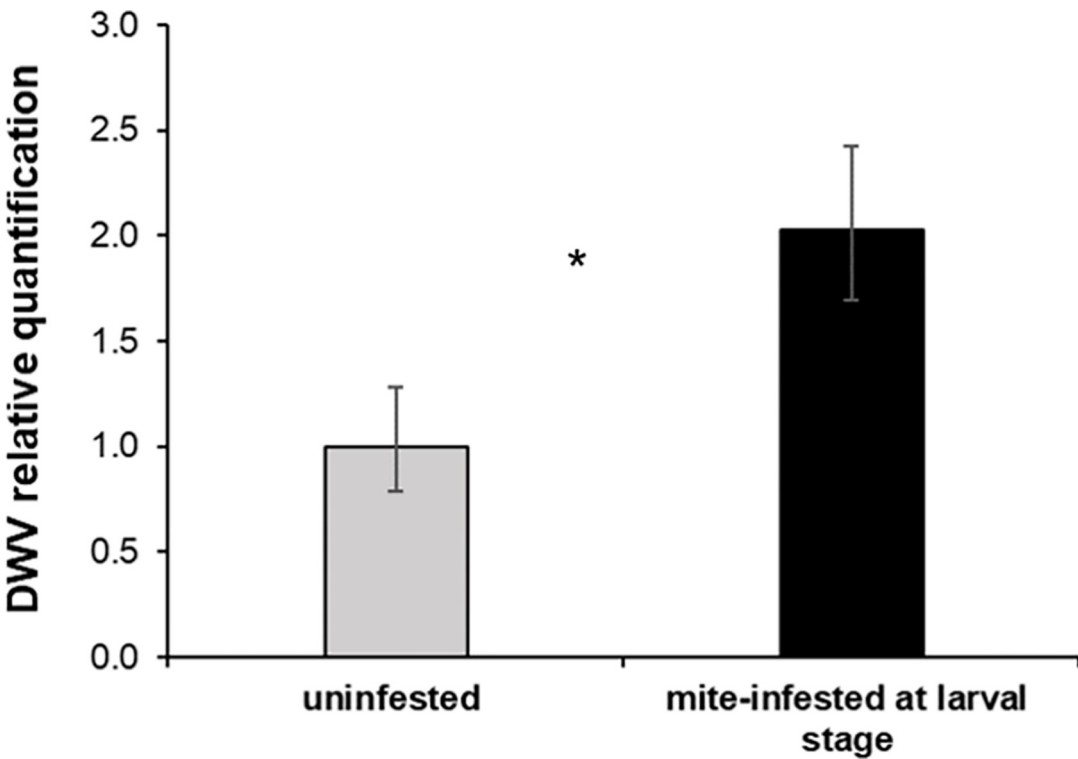

**Fig 4. DWV infection in honey bees mite-infested during the pupal stage.** DWV relative quantification obtained by real time PCR in newly emerged honey bees infested or not with one mite at the L5 stage (Mann-Whitney U test. $U = 22$, $df = 1$, $p = 0.02$, uninfested bees: n = 10, mite-infested bees: n = 10).

The analysis of the DWV genotypes in the bees used for the experiments revealed the presence of two variants: type-A and type-B DWV. A prevalence of reads mapping on type-B DWV was found in all bees regardless of mite infestation and bee's stage at the time of the infestation event (Table 1). The abundance of DWV-A, although generally lower than DWV-B, was not correlated with the host stage. The ratio between the two genotypes in bees infested at different stages (i.e. larvae and newly emerged adults) was similar (Mann-Whitney U test. $U = 24$, $df = 1$, $p = 0.47$).

**2.1.5 DWV copy number in mites.** DWV copy number in *Varroa* mites used for the experiments ranged between 10^3 and 10^9 with most individuals characterized by more than 10^6 viral copies (S1 Fig).

**2.1.6 Effects of mite infestation at different stages on global gene expression of parasitized bees.** To gain insight into the possible causes of the observed differences in survival, we carried out a transcriptomic analysis of bees exposed to mite infestation at different ages.

The comparison between the transcriptome of adult bees that were either mite infested during the pupal stage or not, highlighted 47 differentially expressed genes (FDR < 0.05; S1 Table), two of which were immune-related (LOC724471, 1-phosphatidylinositol 4,5-bisphosphate phosphodiesterase-like and LOC726850, cytochrome b5), while other genes, according to the gene ontology enrichment analysis, were mainly involved in functions as DNA damage response, water transport, cellular amino acid biosynthetic processes, phototransduction and enzymes' activity.

Only four genes were differentially expressed between five-day-old bees infested at the emergence or not; one of them (LOC411577, protein argonaute-2) has an immune-related function being involved in RNA interference (FDR < 0.05, S2 Table).

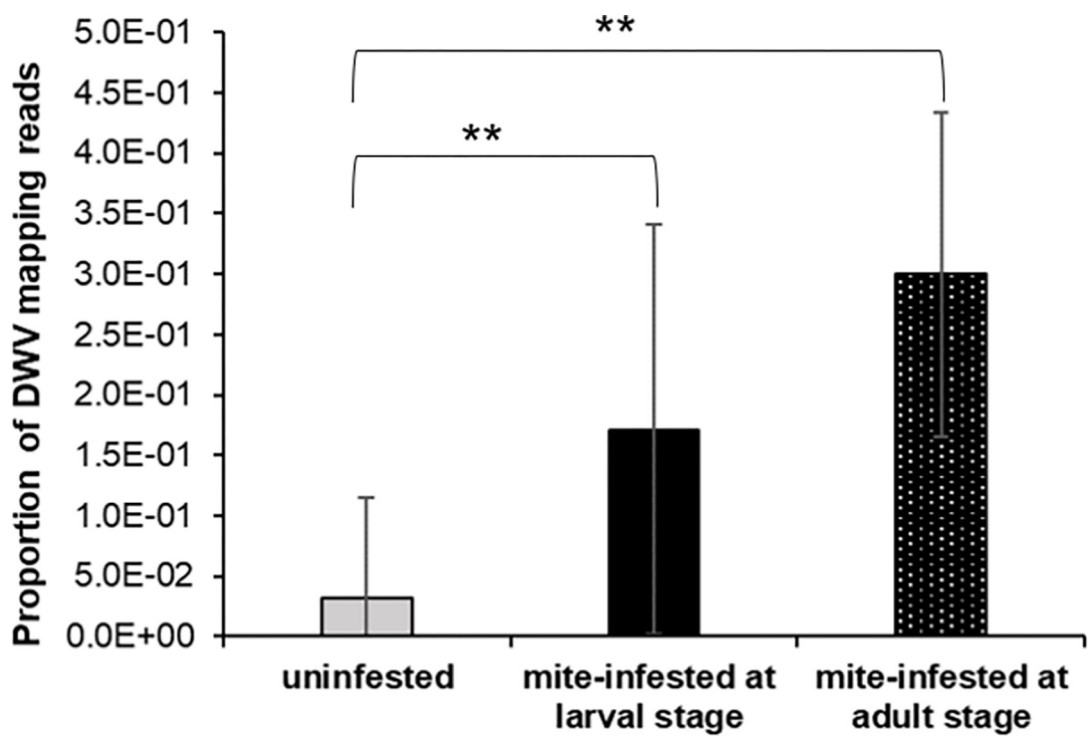

**Fig 5. DWV infection honey bees mite-infested during the pre-imaginal and adult stages.** Proportion of DWV mapping reads in control bees and bees infested at the larval and adult stage (Mann-Whitney U test. Bees infested when larvae vs control: $U = 0$, $df = 1$, $p < 0.01$; bees infested when adults vs control: $U = 0$, $df = 1$, $p < 0.01$, bees infested when larvae vs bees infested when adults: $U = 13$, $df = 1$, $p = 0.07$, uninfested bees: $n = 7$, bees mite-infested at the larval stage: $n = 7$, bees mite-infested at the adult stage: $n = 7$).

Only one gene (LOC726997), a cytochrome b5, appeared to be differentially expressed between bees infested during the pupal stage and after the emergence.

**2.1.7 Effects of mite infestation at different stages on the expression of some selected immune genes.**   Due to their possible role in the response to mite infestation, we assessed, both in pupae and adults artificially infested with one mite or not, the expression of five immune genes: the antimicrobial peptides *Apidaecin* and *Defensin-1*, the transcription factor *Dorsal-1A* and two genes involved in RNA: *Argonaute-2* and *Dicer-like*.

Mite infestation and bees' stage did not affect the expression level of *Apidaecin* (Fig 6A, two-way ANOVA test. Mite: $df = 1$, $F = 0.192$, $p = 0.66$; age: $df = 1$, $F = 1.179$, $p = 0.29$; mite*-age: $df = 1$, $F = 1.237$, $p = 0.28$), although, in mite-infested adult bees, *Apidaecin* appeared to be more expressed than in the other groups. *Defensin-1* was significantly up-regulated in adult honey bees (Fig 6B, two-way ANOVA test. Age: $df = 1$, $F = 294.82$, $p < 0.01$). The same gene was also downregulated in the case of mite infestation (Fig 6B, two-way ANOVA test. Mite: $df = 1$, $F = 9.680$, $p < 0.01$). No significant interaction between the two factors was observed (Fig 6B, two-way ANOVA test. Mite*age: $df = 1$; $F = 0.185$, $p = 0.67$). The gene encoding for

**Table 1. DWV genotypes.**  Percentage of viral reads mapping on the two genomes, type A and B DWV.

|  | uninfested | mite infested when larvae | mite infested when adults |
|---|---|---|---|
| DWV-A | 0.5% | 0.6% | 1.1% |
| DWV-B | 99.5% | 99.4% | 98.9% |

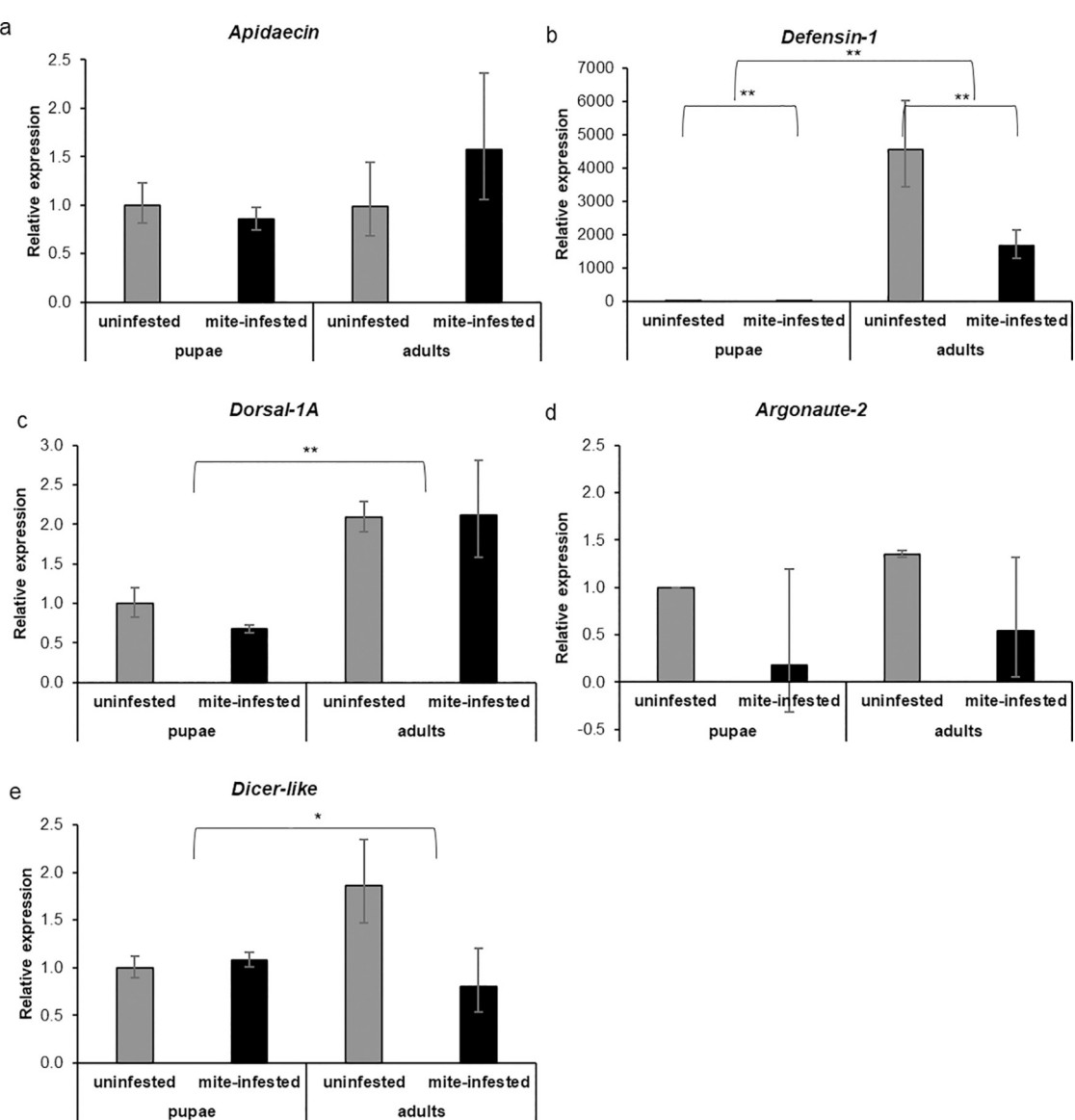

**Fig 6. Gene expression analysis of mite-infested pupae and adult bees.** Relative expression of the immune-related genes *Apidaecin* (a), *Defensin-1* (b), *Dorsal-1A* (c) and of the RNAi related genes *Argonaute-2* (d) and *Dicer-like* (e) of pupae and adult bees challenged with *V. destructor*. (Two-way ANOVA test. *Apidaecin*: mite: $df = 1$, $F = 0.192$, $p = 0.66$; age: $df = 1$, $F = 1.179$, $p = 0.29$; mite*age: $df = 1$, $F = 1.237$, $p = 0.28$; *Defensin-1*: mite: $df = 1$, $F = 9.680$, $p < 0.01$; age: $df = 1$, $F = 294.82$, $p < 0.01$; mite*age: $df = 1$; $F = 0.185$, $p = 0.67$; *Dorsal-1A*: mite: $df = 1$, $F = 2.156$, $p = 0.15$; age: $df = 1$, $F = 37.624$, $p < 0.01$; $df = 1$, $F = 3.058$, $p = 0.09$).; *Ago-2*: mite: $df = 1$, $F = 2.99$, $p = 0.09$; age: $df = 1$, $F = 0.09$, $p = 0.75$; mite*age: $df = 1$, $F = 1.07$, $p = 0.31$,; *Dicer-like*: mite: $df = 1$, $F = 1.90$, $p = 0.18$; age: df = 1, $F = 5.23$, $p = 0.03$; mite*age: $df = 1$, $F = 0.93$, $p = 0.34$. Uninfested pupae: n = 6, mite-infested pupae: n = 6, uninfested adults: n = 6, mite-infested adults: n = 5).

*Dorsal-1A* was significantly up-regulated in adult bees (Fig 6C, two-way ANOVA test. Age: $df = 1$, $F = 37.624$, $p < 0.01$) but was not consistently affected by mite infestation (Fig 6C, two-way ANOVA test. Mite: $df = 1$, $F = 2.156$, $p = 0.15$). No significant interaction was observed between the two factors (Fig 6C, two-way ANOVA test. Mite*age: $df = 1$, $F = 3.058$, $p = 0.09$).

*Argonaute-2* was not affected by the mite (Fig 6D, two-way ANOVA test. Mite: $df = 1$, $F = 2.99$, $p = 0.09$) or even by the host's stage (Fig 6D, two-way ANOVA test. Age: $df = 1$, $F = 0.09$, $p = 0.75$). No interactions were observed between the host stage and mite infestation

([Fig 6D](), two-way ANOVA test. Mite*age: $df = 1$, $F = 1.07$, $p = 0.31$). The RNAi gene *Dicer-like* was significantly affected by the honey bee's stage at the exposure ([Fig 6E](), two-way ANOVA test. Age: $df = 1$, $F = 5.23$, $p = 0.03$) but not by the mite infestation ([Fig 6E](), two-way ANOVA test. Mite: $df = 1$, $F = 1.90$, $p = 0.18$) and no interaction between host stage and parasitization was observed ([Fig 6E](), two-way ANOVA test. Mite*age: $df = 1$, $F = 0.93$, $p = 0.34$).

Overall, most genes appeared to be up-regulated in adult bees, regardless of the mite challenge.

## 2.2 Effects of viral infection on honey bee survival according to the host's life stage

**2.2.1 Effect of viral infection at the larval stage on the survival of bees.** The administration of 10^3 DWV copies to bees at the larval stage significantly reduced their survival once adults ([Fig 7](), Wilcoxon test. Chi square = 6.42, $df = 1$, $p = 0.01$). The percentage reduction in median longevity caused by artificial DWV infection at the larval stage was 39% (median survival: control bees = 18 days; DWV infected bees = 11 days).

The DWV level at the emergence was higher in bees fed with infected larval food but the significance was not reached ([Fig 8](), Mann-Whitney *U* test. $U = 6$, $df = 1$, $p = 0.09$).

**2.2.2 Effect of viral infection at the adult stage on the survival of bees.** The administration of 5 μL of sugar syrup containing 10^3 DWV copies to newly emerged bees caused a significant reduction in adults' survival ([Fig 9](), Wilcoxon test. Chi square = 22.29, $df = 1$, $p < 0.01$). The percentage reduction in median longevity caused by DWV infection at the adult stage was 12% (median survival: control bees = 25 days; DWV infected bees = 22 days).

Although the observed reduced survival, five days after the infection the viral level was similar in the treated and control bees (Mann-Whitney U test. $U = 16$, $df = 1$, $p = 0.37$).

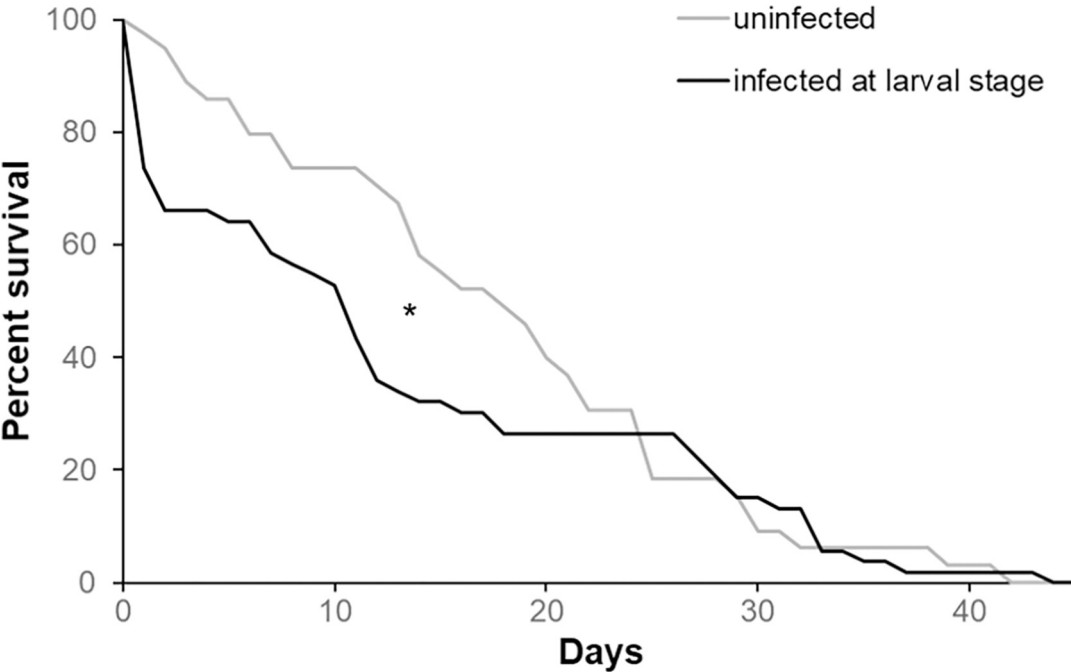

**Fig 7. Survival of the honey bees DWV-infected during the pupal stage.** Survival curves of adult honey bees infected or not with 10^3 DWV copies at the larval stage and maintained under lab conditions (Wilcoxon test. Chi square = 6.42, $df = 1$, $p = 0.01$, uninfected bees: n = 52, DWV-infected bees: n = 50).

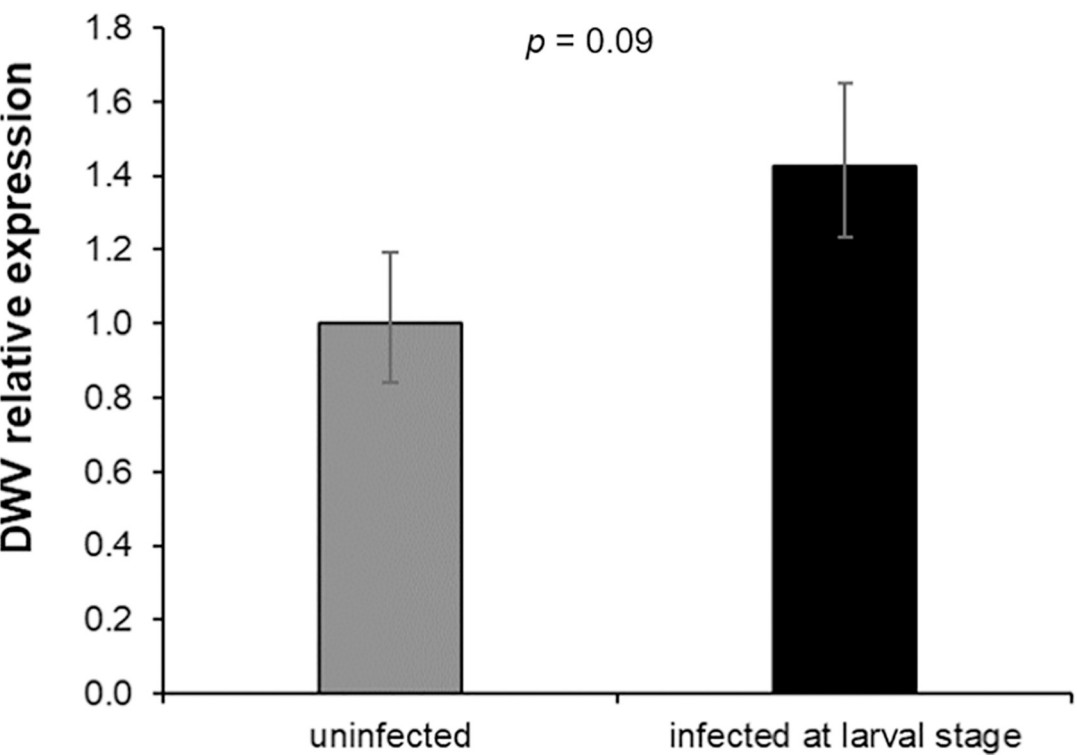

**Fig 8. DWV infection in newly emerged bees DWV-infected during the pupal stage.** DWV relative quantification obtained by real time PCR in newly emerged honey bees infected or not with 10^3 DWV copies at the larval stage (Mann-Whitney U test. $U = 6$, $df = 1$, $p = 0.09$, uninfected bees: n = 5, DWV-infected bees: n = 5).

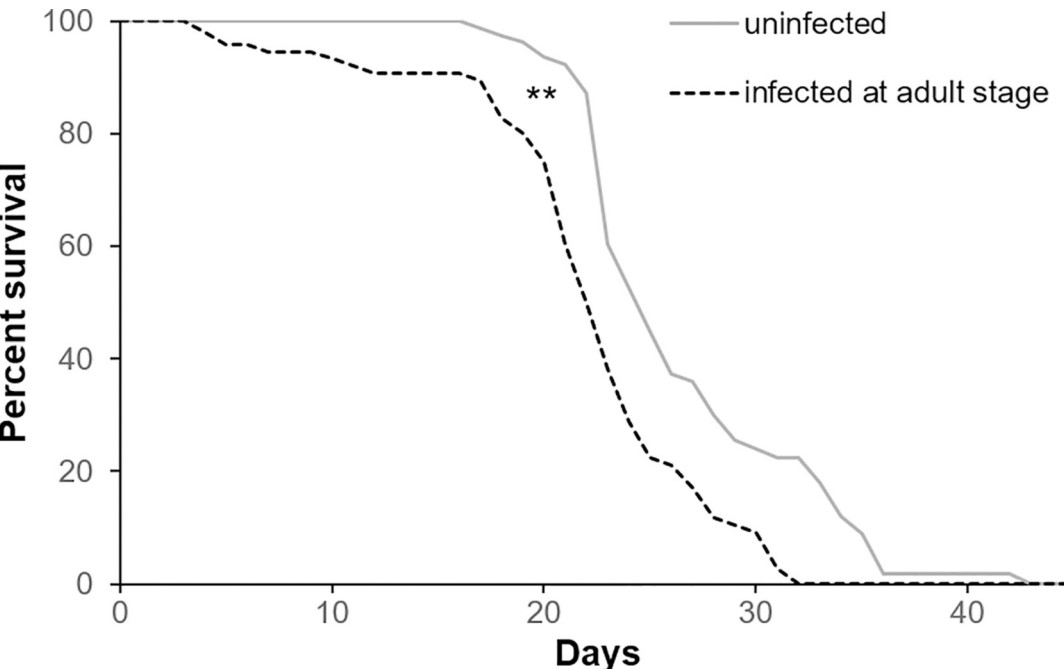

**Fig 9. Survival of the honey bees DWV-infected at the adult stage.** Survival curves of the adult honey bees infected or not with 10^3 DWV copies after the emergence (Wilcoxon test. Chi square = 22.29, $df = 1$, $p < 0.01$, uninfected bees: n = 99, DWV-infected bees: n = 98).

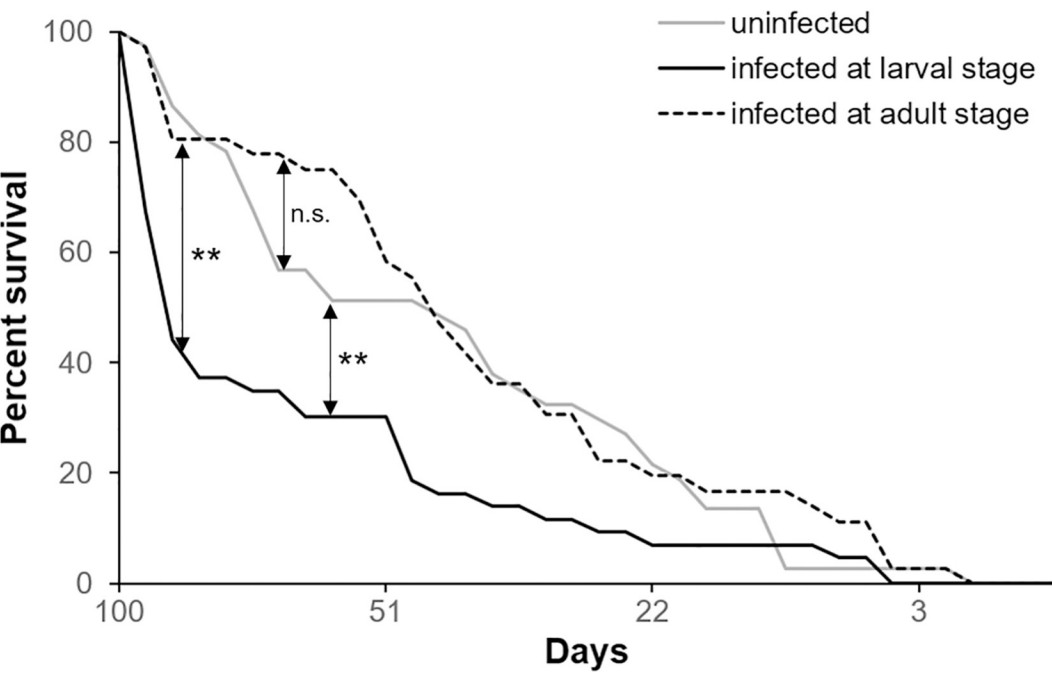

**Fig 10. Effects of the infection's time on bees 'survival.** Survival curves of adult bees infected with 10^3 viral copies at the larval or adult stage (Wilcoxon test. Infection at larval stage vs control: chi square = 14.03, *df* = 1, *p* < 0.01; infection at adult stage vs control: chi square = 0.22, *df* = 1, *p* = 0.63; infection at larval stage vs infection at adult stage: chi square = 14.98, *df* = 1, *p* < 0.1, uninfected bees: n = 37, bees DWV-infected at the larval stage: n = 43, bees DWV-infected at the adult stage: n = 36).

**2.2.3 Effect of the time of viral infection on bees' survival.** The direct comparison of the survival of bees infected with 10^3 viral copies at the larval stage or immediately after the emergence highlighted a reduced survival only if bees were that infected during pre-imaginal stages (Fig 10, Wilcoxon test. Chi square = 14.03, *df* = 1, *p* < 0.01), while the DWV effect on bees infected after the emergence was not confirmed (Fig 10 Wilcoxon test. Chi square = 0.22, *df* = 1, *p* = 0.63). The survival of the adult bees infected at the larval stage was significantly reduced in comparison to that of bees infected after the emergence Fig 10, Wilcoxon test. Chi square = 14.98, *df* = 1, *p* < 0.1).

Similar to mite infestation, DWV infection impacted bee pupae more than adult bees.

**2.2.4 Effects of viral infection at different stages on the expression of some selected immune genes.** Adult bees, both DWV infected and uninfected, showed a higher expression level of *Apidaecin*, *Defensin-1* and *Dorsal-1A* as compared to the pre-imaginal stages. The stage of the bees significantly affected the expression of *Apidaecin* (Fig 11A, two-way ANOVA test. Age: *df* = 1, *F* = 8.43, *p* < 0.01) that was not affected by artificial infection (Fig 11A, two-way ANOVA test. DWV: *df* = 1, *F* = 2.36, *p* = 0.14). No interactions between age and infection were observed (Fig 11A, two-way ANOVA test. DWV*age: *df* = 1, *F* = 1.45, *p* = 0.24). *Defensin-1* was affected both by the host stage and infection: being significantly upregulated in adult bees (Fig 11B, two-way ANOVA test. Age: *df* = 1, *F* = 146.7, *p* < 0.01) and downregulated in presence of DWV infection (Fig 11B, two-way ANOVA test. DWV: *df* = 1, *F* = 4.39, *p* = 0.04). No interactions were observed between the two factors (Fig 11B, two-way ANOVA test. DWV*age: *df* = 1, *F* = 1.00, *p* = 0.32). Finally, the host stage positively affected the expression of *Dorsal-1A* (Fig 11C, two-way ANOVA test. Age: *df* = 1, *F* = 14.47, *p* < 0.01). Instead, this gene was not affected by both viral infection (Fig 11C, two-way ANOVA test. DWV: *df* = 1,

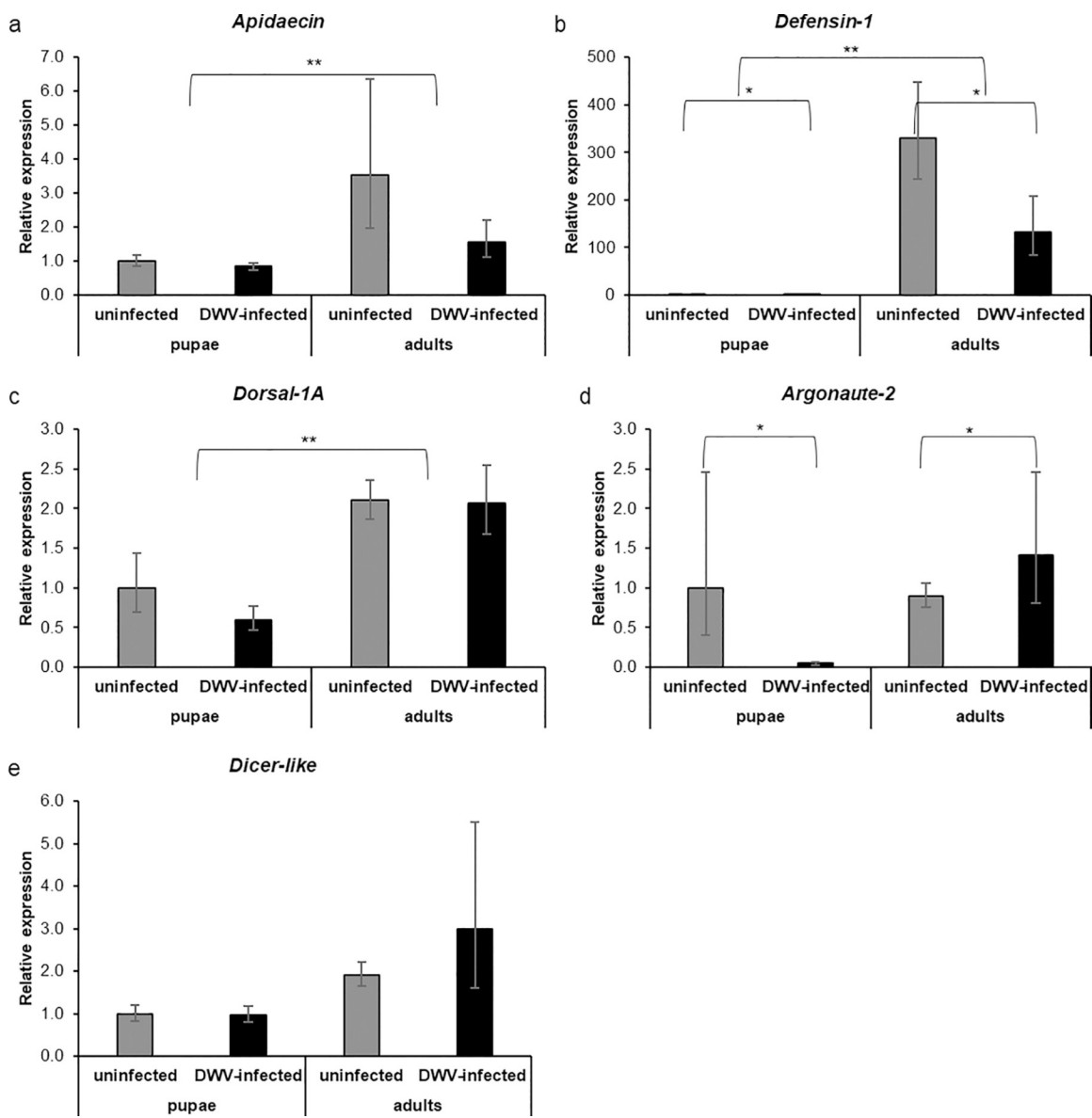

**Fig 11. Gene expression analysis of DWV-infected pupae and adult bees.** Relative expression of the immune-related genes *Apidaecin* (a), *Defensin-1* (b), *Dorsal-1A* (c) and RNA interference related genes *Argonaute-2* (d) and *dicer-like* (e) of pupae and adult bees artificially infected with 10^3 DWV copies. (Two-way ANOVA test. *Apidaecin*: DWV: $df = 1$, $F = 2.36$, $p = 0.14$; age: $df = 1$, $F = 8.43$, $p < 0.01$; DWV*age: $df = 1$, $F = 1.45$, $p = 0.24$; *Defensin-1*: DWV: $df = 1$, $F = 4.39$, $p = 0.04$; age: $df = 1$, $F = 146.7$, $p < 0.01$; DWV*age: $df = 1$; $F = 1.00$, $p = 0.32$; *Dorsal-1A*: DWV: $df = 1$, $F = 1.18$, $p = 0.30$; age: $df = 1$, $F = 4.47$, $p < 0.01$; DWV*age: $df = 1$, $F = 1.01$, $p < 0.32$; *Ago-2*: DWV: $df = 1$, $F = 3.77$, $p = 0.06$; age: $df = 1$, $f = 3.90$, $p = 0.06$; DWV*age: $df = 1$, $F = 7.43$, $p = 0.01$; *Dicer-like*: DWV: $df = 1$, $F = 1.63$, $p = 0.21$; age: $df = 1$, $F = 2.11$, $p = 0.16$; DWV*age: $df = 1$, $F = 1.88$, $p = 0.18$. Uninfected pupae: n = 6, DWV-infected pupae: n = 6, uninfected adults: n = 6, DWV-infected adults: n = 6).

$F = 1.18$, $p = 0.30$) and interaction (Fig 11C, two-way ANOVA test. DWV*age: $df = 1$, $F = 1.01$, $p = 0.32$).

Despite being close to significance, Argonaute-2 was not regulated by host stage (Fig 11D, two-way ANOVA test. Age: $df = 1$, $F = 3.90$, $p = 0.06$) or infection (Fig 11D, two-way ANOVA test. DWV: $df = 1$, $F = 3.77$, $p = 0.06$) but a significant interaction between the two factors was observed (Fig 11D, two-way ANOVA test. DWV*age: $df = 1$, $F = 7.43$, $p = 0.01$). Host's stage

and DWV artificial infection altogether did not significantly affect the expression of Dicer-like (Fig 11E, two-way ANOVA test. Age: $df = 1$, $F = 2.11$, $p = 0.16$; DWV: $df = 1$, $F = 1.63$, $p = 0.21$), although this gene tended to be up-regulated in adult bees. No interaction between host's stage and presence of infection was observed (Fig 11E, two-way ANOVA test. DWV*age: $df = 1$, $F = 1.88$, $p = 0.18$).

Overall, the tested antimicrobial peptides and the upstream transcription factor *Dorsal-1A* appeared to be up-regulated in adult bees, regardless of viral infection, whereas the tested antiviral genes did not respond to both life stage and viral infection.

## 3. Discussion

Parasites represent a major biotic pressure for living organisms but their capacity to harm their host (i.e. virulence) can be affected by several factors. Among these, host age at the exposure is a crucial factor that is often underestimated, especially in invertebrates, despite recent studies have demonstrated how this factor can greatly influence parasite virulence [9].

A number of parasites threat the health of honey bees; some exert their negative impact on a given life stage but a few can impact both the pre-imaginal and the adult stages [44]. Among them, *V. destructor* and its associated virus DWV play a prominent role [45–48]. In fact, this association is responsible for the strong negative pressure exerted on honey bee colonies worldwide [49]. Therefore, understanding how the parasite impacts the different life stages of its host is of great importance for reconstructing the in-hive epidemiology of both the mite and the associated virus. Furthermore, the honey bee, with its social and developmental structure, appears to be a well-suited biological model for testing existing hypotheses regarding the differential virulence of parasites on their host.

Our survival experiments showed that mite parasitization is more detrimental to honey bees if occurring during the pre-imaginal stages, with the higher susceptibility of pupae mirrored by the reduced survival of bees infested as pupae upon reaching adulthood. Mite infestation is normally associated with increasing loads of the Deformed wing virus level [50] and this was confirmed here. However, the higher viral load does not appear to be the primary cause of the observed difference in the survival between bees infested as pupae or adults, because of their similar viral levels. The reduced survival of bees infested during the pupal stage is also unrelated to the DWV variant replicating within the different life stages as revealed by DWV sequencing.

In order to separate the effect of the mite from that of the vectored virus, we artificially infected bee larvae at the 4th instar or immediately after the emergence with a DWV load comparable to that transmitted by the mite. Again, adult bees' survival was significantly reduced in bees infected during the larval stage, while the infection after the eclosion had little if any effect on survival.

In sum, both the mite and the virus differentially impacted the host according to the life stage at the time of exposure, suggesting that the observed higher parasites' negative effects on the pre-imaginal stages is related to some age-dependent host features rather than to the intrinsic characteristics of the parasite itself and the associated virus.

Transcriptomic analysis of five-day-old adult bees infested with one mite at different life stages did not highlight any specific gene expression pattern involving genes related to stress response and the immune system. Consequently, the analysis of the individual-level life stage effects was based on comparing the expression levels of some immunity-related key genes between honey bees' pupae and adults immunostimulated or not with the mite and DWV.

The expression of a gene encoding for the kB nuclear factor *Dorsal-1A* and two antimicrobial peptides (i.e. *Apidaecin* and *Defensin-1*) appeared to be different between pupae and

adults, both in case of mite infestation and viral challenge, with pupae showing a reduced expression of such genes in comparison to adults. Of the three selected immune genes, only *Apidaecin* was not significantly down-regulated in pupae and only in one of the two parasitic challenges investigated here.

In the case of the two genes implicated in RNAi (i.e. *Argonaute-2* and *Dicer-like*), the effect of the host life stage at exposure was not as clear as above. Indeed, only the *Dicer-like* expression in mite-infested honey bees, followed the same pattern of above. However, the viral load administered to pupae may have not triggered a response.

In general, in honey bee pupae, a lower expression of immune genes involved in the humoral and antiviral response was observed here. This evidence, confirmed using both a parasitic mite and a virus, is indicative of a reduced immune competence of bee pupae as compared to adults.

Our results are in line with previous evidence reported by Gatschenberger and collaborators [39] in a study on the cellular and humoral defense reactions through three different bee life stages: larvae, pupae and adults. In the described experiments, pupae artificially infected with low doses of *E. coli*, in contrast to larvae and adults, were unable to mount an effective cellular or humoral reaction, showing no fast clearance of viable bacteria and no induction of AMPs. In line with what has been observed by Gatschenberger and collaborators [39], our results indicate that pupae represents the most vulnerable stage during honey bee development.

Overall, these results seem to confirm the existence of a trade-off between immunity and growth, such that bee pupae, endowing a drastic remodelling during the last metamorphose would do so at the expense of an effective defence against parasites.

It is worth noting that the reduced immune competence of pupae is not related to the absence of an immune-reactive tissue. Actually, that stage is characterized by a high concentration of hemocytes [51] and a functional fat body [52] that are both central for immunity. However, the activation of an efficient immune response requires a large energetic investment which could not be maintained during the metamorphosis, when most of the larval tissues are recycled and remodelled to support the re-architecture of the animal into the adult [40].

The apparent vulnerability of a crucial developmental stage in the life of bees may represent a serious handicap whose existence needs to be justified. In fact, honey bees can rely on a further level of immunity: the so-called social immunity [13] which includes a suite of behaviors that complement individual defense. Interestingly, social immunity seems to be particularly effective to protect bee pupae. Indeed, the enclosure of the developing bee into a protected environment preventing any potentially dangerous interactions may well compensate for the lack of the capacity to react against possible infectious contacts. Unfortunately, the recent introduction of the Asiatic bee mite *V. destructor* that, after slipping into a brood cell before sealing, can cause, together with the vectored DWV, significant damage to the honey bee, taking advantage of its reduced immunocompetence, made the confinement of the brood into a protected cell a less efficient strategy. Interestingly, mite-infested colonies seem to be able to display a further level of defense based on the application of propolis to the brood cells in order to limit mite's survival and reproduction [53], indicative of an ongoing tough arm-race between the honey bee and its parasites.

In conclusion, our study highlighted a stage-dependent susceptibility of honey bees to the mite *V. destructor* and the deformed wing virus, an achievement which will help to better understand the impact of those parasites on honey bees and colony health. Further experiments are needed to strengthen the hypothesis of a trade-off between immunity and development but the reduced expression of immune-related genes showed here by honey bee pupae seems to support this interpretative framework. This, in turn, could allow further

advancements in the elucidation of the factors affecting the virulence of invertebrate parasites and consequently allow to gain a better insight into their biology, effects and practical consequences.

## 4. Material and methods

### 4.1 Biological materials

**4.1.1 Honey bees and *Varroa* mites.** Honey bees and *Varroa* mites used in all the experiments were collected from the experimental apiary of the Dipartimento di Scienze AgroAlimentari, Ambientali e Animali of the University of Udine (Udine, Italy 46˚04'54.2" N, 13˚12'34.2" E). Previous studies indicated that local colonies are hybrids between *A. mellifera ligustica* Spinola and *A. mellifera carnica* Pollmann [54, 55].

**4.1.2 Deformed wing virus.** Deformed wing virus particles were isolated and purified by ultracentrifugation from eight groups of five symptomatic bees collected from the apiary of the University of Udine, following step by step the protocol described by De Miranda et *al.* [56]. In this way, eight extracts were obtained and maintained in PBS buffer 0.1 M, at 4˚C until use. Each DWV extract was analyzed at the Agricultural Science Department of the University Federico II of Naples and the number of viral particles quantified according to the protocol described by Di Prisco et *al.* [25]. There the extract was also analyzed by real time PCR using convenient primers [57–59] in order to exclude the presence of three other common bee viruses: Kakugo virus, Sacbrood virus and Black queen cell virus (F. Pennacchio, personal communication).

**4.2 DWV quantification in mites.** Ten individual mites were homogenised under liquid nitrogen into a 2 mL microcentrifuge tube with a plastic micro pestle. Total RNA was extracted from each mite adopting TRIzol® and chloroform extraction combined with RNAeasy® mini spin column (Qiagen, Germany) following the protocol described by the Untergasser's lab [60]. The amount of total RNA was quantified by means of a Nanodrop® spectrophotometer. cDNA was synthesized starting from 100 ng of RNA following the manufacturer specifications (M-MLV reverse transcriptase, PROMEGA Italy). 10 ng of cDNA from each sample were analyzed in triplicate by qRT-PCR using SYBRgreen dye on a BioRad CFX96 Touch™ detector. The following thermal cycling profiles were adopted: one cycle at 95˚C for 10 minutes, 40 cycles at 95˚C for 15s and 60˚C for 1 min, and one cycle at 68˚C for 7 min. DWV primers are reported in Table 1. DWV copy number was determined by plotting the Ct values of the unknown samples to an established standard curve obtained by plotting the logarithm of eight 10-fold dilutions of a starting solution containing 0.5 ng/μL of synthetic DWV copies (gBlocks, IDT, US) against the corresponding Ct value. The PCR efficiency (102.7%) was calculated according to the formula $E = 10(-1/slope) - 1$ (slope $= -3.260$, y-intercept $= 2.60$, $R2 = 0.99$).

### 4.3 Artificial infestation with *Varroa destructor* mite

**4.3.1 Larvae.** L5 honey bee larvae were obtained from brood cells capped in the preceding 15 h, as described by Nazzi and Milani [61]; bee larvae were then artificially infested with one mite or maintained uninfested as controls and put into 6.5 mm i.d. gelatin capsules (Agar Scientific Ltd, UK). Since the feeding activity of the mites only starts six hours after cell sealing [62], by using this method, we could exclude that bees classified as uninfested had been infested prior to the experiment.

Ten mites obtained from the brood cells were immediately killed with liquid nitrogen and stored at -80˚C in order to quantify the DWV in the parasite vector. Mite-infested and control bees were maintained for 12 days at 34.5˚C, 75% RH, dark [61]. At eclosion, the bees were

cleaned from the mites, transferred into plastic cages (185 × 105 × 85 mm), fed with sugar candy (Apifonda®) and water ad libitum and maintained at 34.5˚C, 75% RH. Six newly emerged uninfested and infested bees were sampled and stored at -80˚C for the viral quantification. The experiment was replicated three times; about 150 bees per treatment were encapsulated. At the end of the development 103 uninfested bees and 76 artificially mite-infested bees have survived and been caged.

**4.3.2 Adult bees.**   The day before the experiment, several brood combs containing bees ready to emergence were randomly collected from the experimental apiary and stored overnight in a climatic chamber (34.5˚C, 75% R.H. dark). The day after, newly emerged bees were infested with one mite collected from the same comb, or maintained uninfested as control. In principle, adult bees emerging from brood cells could be infested at the pupal stage, however, given the infestation rate of brood cells in the hives used for the experiment when the trial was carried out (about 20 mites/1000 bees), this was likely true only for a small minority of bees. In any case, all the bees used in the experiment were carefully inspected before the artificial infestation and, at eclosion, the mite tends to stay on the body of the emerging bees (often hiding in the petiole area) [63] and it should be noted through a careful visual inspection.

The accidental detachment of the mites from the infested adult bees and their movement from one bee to another, can result in a little portion of bees being uninfested for some time and others suffering the effects of a double infestation. To make sure that this would not affect our results we carried out a preliminary trial in which adult bees were infested with one or two mites. Since no differences were observed, in terms of survival, between the honey bees of the two groups (Wilcoxon test = 0.477, $df$ = 1, $p$ = 0.49) we considered the possible fall of infesting mites and the occasional double infestation due to mites' movement as a seldom un-influent possibility with respect to the assessment of the effects of mite infestation on bee survival.

Adult bees were transferred into plastic cages and maintained under lab conditions as described above. After five days, six uninfested and infested bees were sampled and kept at -80˚C for the viral quantification.

The experiment was replicated three times using in total 82 uninfested bees and 83 artificially mite-infested bees.

## 4.4 Mite effects according to the honey bees' life stage

In order to compare the mite's effects according to the honey bee's life stage, honey bee larvae were artificially infested at the 5th instar with one or no mites as described above. Twelve days later, at eclosion, part of the uninfested newly emerged adults were artificially infested with one mite collected from the bees infested during the larval stage. All the bees were caged, fed with water and sugar candy *ad libitum* and maintained at 34.5˚C, 75% R.H. dark; seven bees per experimental group were sampled after five days for RNA sequencing.

The experiment was repeated three times; about 200 bees per treatment were encapsulated. At the end of development 164 uninfested bees, 140 bees artificially mite-infested as larvae and 121 artificially mite-infested bees infested at the emergence have survived and been caged.

All the caged bees used for the above-described experiments were daily monitored and the number of dead individuals was recorded.

## 4.5 RNAseq analysis

The whole body of seven bees per thesis (*i.e.* uninfested, mite-infested during the pupal stage and mite-infested at the adult stage) was homogenized by means of mortar and pestle in liquid nitrogen. Total RNA was extracted and purified according to the procedure provided with the RNeasy Plus mini kit (Qiagen®, Germany). The amount and the integrity of the RNA in each

sample were quantified by means of a Lab chip GX touch nucleic acid analyzer (Perkin Elmer ™ UK). Libraries preparation and RNA sequencing in paired reads of length 150 bp were performed by IGA technology services s.r.l. of Udine (Italy) using a NovaSeq™ platform (Illumina, US).

To simultaneously quantify the abundance of bee transcripts and viral particles, reads were aligned using STAR [64] on a reference genome composed of the genomes of *Apis mellifera* (RefSeq accession GCF_003254395.2), DWV-A (GCF_000852585.1) and DWV-B (GCF_000856945.1), setting the quantMode parameter to "GeneCounts", to obtain the number of reads mapping to each transcript.

Differential expression was assessed using the R package DESeq2 [65].

## 4.6 Immune response in mite-infested pupae and adults

In order to evaluate the different immune response between pupae and adults infested by the mite, 5th instar larvae were artificially infested with one mite or maintained uninfested as above described. Seven purple eyed-pupae were killed in liquid nitrogen and put in a -80˚C refrigerator for the gene expression analysis. After twelve days, parts of the newly emerged uninfested bees were artificially infested with one mite or maintained uninfested as control, caged and sampled when five-day-old adults.

## 4.7 Artificial infection with DWV

Previous studies [66, 67] suggest that mites harbouring viral loads > 10^8 DWV genome equivalents can transmit to bees about 2500 viral copies (representing the $OID_{50}$, overt infection dosage). Furthermore, the viral load in commercial honey and pollen ranges from 10^2 to 10^6 per g [29, 68] while there are no accurate data regarding DWV copy number in the royal jelly, although the virus has been identified in the secretions of the hypopharyngeal glands and in the larval food [28, 69]. Therefore, in order to compare the effects of *Varroa* infestation and the feeding upon contaminated food on viral infection, we administered to each bee 10^3 viral copies by food.

**4.7.1 Larvae.** Brood combs hosting 4th instar larvae were collected from the experimental apiary. Larvae, gently extracted from the brood cells, were transferred into a 96-well ELISA microplate filled with 40 mg of a larval diet consisting of gamma-rays (25 kgy) sterilized royal jelly, glucose, fructose and yeast extract [70]. The artificial diet in every well was supplemented with 1 ul of PBS buffer 0.1 M containing 10^3 DWV copies. An equivalent number of larvae was fed with the same diet containing the buffer devoid of viral particles. The day after, at the reaching of the 5th instar larval stage, bees were transferred into Petri dishes and maintained at 34.5˚C, 75% R.H. for 12 days to complete the development. After the eclosion, six newly emerged infected bees and six uninfected individuals were sampled and put in a -80˚C refrigerator for viral quantification. The remaining bees were caged and fed with water and sugar candy *ad libitum* as described in the paragraph 4.3.

The experiment was repeated three times; about 150 four instar larvae per treatment were prepared.

Due the high endling involved in this type of experiment, at the end of the development 52 uninfected bees and 50 artificially infected bees have survived and been caged.

**4.7.2 Adult bees.** Brood combs with eclosing bees were collected from the experimental apiary and put in a climatic chamber at 34.5˚C, 75% R.H., dark for one night. The day after, newly emerged bees were fed with 5 μL of sugar syrup added with 1000 DWV copies. Bees were fed one by one using a micropipette (Pipetman®) equipped with a 10 μL tip; to be sure that bees ate syrup, they were starved for one hour and stimulated to extend the ligula by gently

touching the antennas with a drop of syrup. Part of bees was fed with uncontaminated syrup as control. After feeding, bees were immediately caged and maintained as above described. After five days, six bees were sampled and put in -80˚C refrigerator for the viral quantification.

The experiment was repeated three times using in total 99 uninfected bees and 98 artificially infected bees.

## 4.8 DWV effects according to the honey bees life stage

In order to directly compare DWV effects according to the honey bee's age, bees were fed with 10^3 viral copies when 4th instar larvae or maintained uninfected as controls. Twelve days later, after the emergence, part of the uninfected newly emerged adults was fed with 5 μL of sugar syrup containing 10^3 viral copies. All the bees were caged, fed with water and sugar candy ad libitum and maintained at 34.5˚C, 75% R.H., dark.

All the experiments have been replicated three times; about 150 four instar larvae per treatment were prepared. Due the high endling involved in this type of experiment, 37 uninfected bees, 43 bees artificially infected as L4 larvae and 36 bees artificially infected after the emergence have survived and been caged.

All the caged bees used for the above described lab experiments were daily monitored and the number of death and deformed individuals recorded. The experiments were carried out from the middle of June to the middle of July, when mite infestation and viral levels in the Northeast of Italy are still low (in the order of 20 mites/1000 bees) but there are enough *Varroa* mites to conduct the experiments.

## 4.9 Immune response in infected pupae and adults

In order to find possible differences in the immune response between pupae and adults, 4th instar larvae were artificially infected via food with 10^3 viral copies or maintained uninfected as control. Seven developing bees were killed in liquid nitrogen as purple eyed pupae and stored at -80˚C for subsequent analysis. After twelve days, part of the newly emerged uninfected bees was artificially infected with 10^3 viral copies administered with 5 μL of sugar syrup or maintained uninfected as control. Bees were caged and sampled five days after the emergence.

## 4.10 qRT-PCR analysis

The whole body of six bees was homogenized by means of mortar and pestle in liquid nitrogen. Total RNA was extracted and purified according to the procedure provided with the RNeasy Plus mini kit (Qiagen®, Germany). cDNA was synthetized starting from 500 ng of RNA following the manufacturer specifications (M-MLV reverse transcriptase, PROMEGA Italy). Additional negative control samples containing no RT enzyme were included. 10 ng of cDNA from each sample were analyzed using qRT-PCR with the primers reported in Table 2 using SYBRgreen dye (Ambion®), according to the manufacturer specifications, on a BioRad CFX96 Touch™ Real time PCR Detector. In order to ensure that primers efficiency was included in the desirable range of 90–110%, this was calculated according to the formula $E = 10^{(-1/slope)} - 1) * 100$. The following thermal cycling profiles were adopted: one cycle at 95˚C for 10 minutes, 40 cycles at 95˚C for 15s and 60˚C for 1 min, and one cycle at 68˚C for 7 min. Relative quantification of genes and DWV was performed adopting with the 2−ΔΔCt method [71] using actin as the housekeeping gene after having verifying its stability in the pre-imaginal and adult bees stages according to de Jonge et al. [72] (coefficient of variation below 5% and ratio of the maximum and minimum values observed within the dataset < 2).

**Table 2. Real time PCR primers.** Sequences of the primers used for real time PCR analysis.

| Name | Sequence | Reference |
| --- | --- | --- |
| *Actin*, forward | GATTTGTATGCCAACACTGTCCTT | [25] |
| *Actin*, reverse | TTGCATTCTATCTGCGATTCCA | |
| DWV, forward | GGTAAGCGATGGTTGTTTG | [73] |
| DWV, reverse | CCGTGAATATAGTGTGAGG | |
| *Apidaecin*, forward | TTTTGCCTTAGCAATTCTTGTTG | [74] |
| *Apidaecin*, reverse | GAAGGTCGAGTAGGCGGATCT | |
| *Defensin-1*, forward | TGCGCTGCTAACTGTCTCAG | [35] |
| *Defensin-1*, reverse | AATGGCACTTAACCGAAACG | |
| *Dorsal-1A*, forward | ACAGGCAGAAGCTGAGAAGC | [75] |
| *Dorsal 1A*, reverse | TTGCCATCGGATACAAGGAT | |
| *Ago-2*, forward | AAAAAGAGCTATTGCGCGCT | [36] |
| *Ago-2*, reverse | ACCATAACTCGGAGCGTTGA | |
| *Dicer-like*, forward | TGCAGAATGAATCAAAGAACCGA | [76] |
| *Dicer-like*, reverse | TGAGCCAATACAAAGCTGGA | |

## 4.11 Statistics

Each survival curve was compared with its control using the Wilcoxon test after checking for validity of the proportional hazard (PH) assumption by means of R software version 4.1.3 [77]. Since in several cases the hazard ratio was non-constant, the generalized Wilcoxon test appeared more suitable for the survival curves' comparison [78]. All the survival curves obtained from the experiments with caged bees were analyzed with Minitab®.

The relative quantities of DWV in the treated and control individuals and the gene expression data were analyzed by means of R software version 4.1.3 [77] adopting a Mann-Whitney U test and a two ways ANOVA respectively.

## Supporting information

**S1 Fig. DWV copy number in mites.** Log transformed viral copy number in ten individual mites collected from the experimental apiary of the University of Udine during the period when the experiments were carried out.
(DOCX)

**S1 Table. DEGs bees infested during the pre-imaginal stages.** List of the genes differentially expressed between adult bees infested during the larval stage and uninfested.
(DOCX)

**S2 Table. DEGs bees infested at the adult stage.** List of the genes differentially expressed between adult bees infested after the emergence and uninfested.
(DOCX)

## Acknowledgments

We gratefully acknowledge Gammatom Srl (Como, Italy) for performing for free the sterilization with gamma rays of the royal jelly used in this study.

## Author Contributions

**Conceptualization:** Virginia Zanni, Desiderato Annoscia, Francesco Nazzi.

**Data curation:** Virginia Zanni, Davide Frizzera, Fabio Marroni, Elisa Seffin, Desiderato Annoscia, Francesco Nazzi.

**Formal analysis:** Fabio Marroni.

**Investigation:** Virginia Zanni, Davide Frizzera, Elisa Seffin, Desiderato Annoscia, Francesco Nazzi.

**Writing – original draft:** Virginia Zanni, Francesco Nazzi.

**Writing – review & editing:** Virginia Zanni, Desiderato Annoscia, Francesco Nazzi.

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
