## [Decision Letter · Decision Letter 0]

2 May 2023

PONE-D-23-04947Age-related response to mite parasitization and viral infection in the honey bee 1 suggests a trade-off between growth and immunityPLOS ONE

Dear Dr. Zanni,

Thank you for submitting your manuscript to PLOS ONE. After careful consideration, we feel that it has merit but does not fully meet PLOS ONE’s publication criteria as it currently stands. Therefore, we invite you to submit a revised version of the manuscript that addresses the points raised during the review process.

As you can see, both reviewers liked your study in general but have raised a number of concerns that should be addressed.

We look forward to receiving your revised manuscript.

Kind regards,

Olav Rueppell

Academic Editor

PLOS ONE

Journal Requirements:

"European Union's Horizon 2020 research and innovation 639 program, under Grant Agreement No. 773921 (PoshBee) and the Italian Ministry of University, PRIN 640 2017 - UNICO (2017954WNT)."

"no potential conflict of interest was reported by the authors."

Reviewers' comments:

Reviewer's Responses to Questions

**Comments to the Author**

1. Is the manuscript technically sound, and do the data support the conclusions?

Reviewer #1: Yes

Reviewer #2: Yes

2. Has the statistical analysis been performed appropriately and rigorously? 

Reviewer #1: Yes

Reviewer #2: Yes

3. Have the authors made all data underlying the findings in their manuscript fully available?

Reviewer #1: No

Reviewer #2: Yes

4. Is the manuscript presented in an intelligible fashion and written in standard English?

Reviewer #1: Yes

Reviewer #2: Yes

5. Review Comments to the Author

Reviewer #1: The manuscript by Zanni et al. presents a range of experiments on mite parasitism of different honey bee like stages, and experimental inoculations with DWV to investigate trade-offs between immunity and growth. The study is nicely completed with gene expression analyses. Although the results are definitely worth publishing, I have a few comments that need addressing and suggestions outlined below.

One concern is in regards to the nature of viral challenge used in the infection protocols. Not enough details are given in the methods on how the inoculum was prepared, especially whether it was the same inoculum throughout the study. Line 470, the authors write "the presence of additional viruses besides DWV has been ruled out" but no information is given on how they did that (qPCR? sequencing?). That is especially relevant if there were more than one inoculum prepared. Please give additional information on these points.

In Section 4.3, it is unclear how the authors controlled for mite infestation prior to experimental infestation. How do we know, for instance, that "uninfected bees" did not get infested and lost their phoretic mite after emerging? If the frame is from a regularly treated and varroa-monitored hive, adding some data on this is necessary (e.g. mite counts, or less preferably at least varroa treatments timing). Alternatively, the authors could rule out prior mite infestations if they did not find any signs of varroa infestation in the cells of the newly emerged workers.

Some of the immune genes are down regulated in infested bees - could that be a case of immunosuppression that is often discussed in the Varroa-DWV symbiotic relationship? I am aware that the study was not designed with that intent and that the data presented might not be enough to support strong claims on the matter, but I think it is a valuable element of discussion that could be worth including. However, I will leave that decision to the authors, and will accept if they disregard my comment.

Related to the gene expression analysis - is it correct to compare gene expression between pupae and adults using only one reference gene? Has the reference gene been validated to be constantly expressed between the pupae and imago? Please provide a reference, or some form of data that indicates actin as a suitable reference gene. If no data or reference is available, please add a disclaimer so readers are aware of this problem.

The paragraph lines 506-511 is unclear and needs rephrasing: I understand that the difference between infestation by 1 vs 2 mites is negligible, but I don't see how it justifies the effect of "accidental fall of the mites". Please give more rationale.

Throughout the paper, sample size should be given more clearly for each experiment/plot. I'd suggest adding points to the barplots to get a better idea of data distribution (using function geom_points() in ggplot2 for instance), but the authors can choose to not do that and instead at least adding sample size in relevant plot captions. This is partly why I've ticked "no" in the review form question "Have the authors made all data underlying the findings in their manuscript fully available?". It is also essential that you deposit your raw data on a public repository (e.g. NCBI's SRA). You should also review how you report statistical tests, i.e. italics for things like "df", "p", spaces around "=" signs, etc

Line 226-227: this statement needs more details, e.g. are you able to plot the distribution of viral loads in Varroa for the samples you characterised? At least give median, mean and standard deviation to be more descriptive.

Line 287, this statement needs a reference. I'm not sure how the artificial feeding of DWV relates to natural inoculation via mite parasitism in terms of quantity of virus transmitted to the bees.

Line 375, replace "nice" with something less colloquial, e.g. "well suited"

Line 405, what is meant by "regardless of the clear response to the parasitic challenge"? Please be more specific.

Line 428: replace "isn't" by "is not".

Line 561-563, this statement appears too vague to be relevant (and is not referenced). What do the authors mean by "the contribution likely offered by the mite"? I'd think that natural viral inoculation by Varroa is variable and highly dependant on the mites viral load, sure, but also duration of feeding on the bee, etc.

In section 4.8 line 588 (and anywhere else relevant), please give the inoculum volume given to larvae in addition to viral titre. is it 5 uL as for the adult bees?

Reviewer #2: This is a well-written article about the importance of Varroa infestation and DWV infection at different life stages of honey bee. The experimental design was simple but adequate. The results were clear and easy to understand. The discussion was supported by the results and integrated with previous findings. Overall, a high quality manuscript. However, there are a few minor comments that should be easily addressed before publication.

167 How was Varroa infestation during the pupal stage controlled for in experiment on adult infestation? I'm sure it was part of the protocol but I couldn't find anything about how Varroa were managed at the colony level to make sure there was no impact of pupal infestation on results of the adult bee infestation with Varroa.

199 If the difference is not significant, then this sentence should read "There was no significant difference in DWV level in bees infested with Varroa and the control" or something to that effect.

460 What time of year was this study performed? This is important as Varroa populations and viral dynamics in source colonies can significantly influence the outcomes of the experiments.

502 This is a call back to the comment on line 167. How was Varroa infestation in the pupal stage controlled for in the experiment on infesting adult bees with Varroa?

All the figures were simple, but very effective at showing the data. However, adding indicators of significant differences would make the data much easier to understand visually without having to read the figure legend. This would improve the figures that have multiple treatment groups such as Figures 5, 6, 10, and 11.

6. PLOS authors have the option to publish the peer review history of their article (what does this mean?). If published, this will include your full peer review and any attached files.

Reviewer #1: No

Reviewer #2: No

---

## [Author Response · Author response to Decision Letter 0]

30 Jun 2023

Dear Editor and Dear Reviewers,

Thank you for the helpful comments concerning the manuscript “Age-related response to mite parasitization and viral infection in the honey bee suggests a trade-off between growth and immunity”. 

We have addressed each comment in the revised version of our manuscript. Please find below (in plain) the list of the comments and, in italics, our response to the Reviewers.

Sincerely,

 The Authors 

Reviewer 1

The manuscript by Zanni et al. presents a range of experiments on mite parasitism of different honey bee like stages, and experimental inoculations with DWV to investigate trade-offs between immunity and growth. The study is nicely completed with gene expression analyses. Although the results are definitely worth publishing, I have a few comments that need addressing and suggestions outlined below. 

We thank reviewer 1 for her/his positive assessment of our work.

R1.1: one concern is in regards to the nature of viral challenge used in the infection protocols. Not enough details are given in the methods on how the inoculum was prepared, especially whether it was the same inoculum throughout the study. Line 470, the authors write "the presence of additional viruses besides DWV has been ruled out" but no information is given on how they did that (qPCR? sequencing?). That is especially relevant if there were more than one inoculum prepared. Please give additional information on these points.

Thank you for the observation and the suggestion.

We isolated the DWV from symptomatic bees and used the same inoculum throughout the study. We followed step by step the protocol described by De Miranda and collaborators in the article “Standard methods for virus research in Apis mellifera”, paragraph number 7 (Journal of Apicultural Research52(4): http://dx.doi.org/10.3896/IBRA.1.52.4.22). 

Since the protocol is quite long we decided to report just the bibliographic reference. The DWV extract was sent to our collaborator Professor F. Pennacchio at the Agricultural Science department of the University Federico II (Naples, Italy) where the DWV in the extract was quantified adopting the method described in the supporting information of Di Prisco et al., 2016 (A mutualistic symbiosis between a parasitic mite and a pathogenic virus undermines honey bee immunity and health, https://doi.org/10.1073/pnas.1523515113). They also excluded the presence of other three main bee viruses: Kakugo, Sacbrood and Black queen cell viruses.

To add this important piece of information to our manuscript, we reformulated the previous sentence (Lines 467-470 of the original document) as follow: “Deformed wing virus particles were isolated and purified by ultracentrifugation from eight groups of five symptomatic bees collected from the apiary of the University of Udine, following step by step the protocol described by De Miranda et al. [56]. In this way, eight extracts were obtained and maintained in PBS buffer 0.1 M, at 4 °C until use. Each DWV extract was analyzed at the Agricultural Science Department of the University Federico II of Naples and the number of viral particles quantified according to the protocol described by Di Prisco et al. [25]. There the extract was also analyzed by real time PCR using convenient primers [57–59] in order to exclude the presence of three other common bee viruses: Kakugo virus, Sacbrood virus and Black queen cell virus (F. Pennacchio, personal communication).” Lines 480-487 of the revised document.

R1.2: In Section 4.3, it is unclear how the authors controlled for mite infestation prior to experimental infestation. How do we know, for instance, that "uninfected bees" did not get infested and lost their phoretic mite after emerging? If the frame is from a regularly treated and varroa-monitored hive, adding some data on this is necessary (e.g. mite counts, or less preferably at least varroa treatments timing). Alternatively, the authors could rule out prior mite infestations if they did not find any signs of varroa infestation in the cells of the newly emerged workers.

Honey bee larvae were collected from brood cells that were sealed in the preceding 15 hours. To do so, the evening before the experiment, we marked all the sealed cells of a convenient number of brood combs containing enough mature bee larvae. The day after, we transferred the marked combs to the lab, removed the sealing of the unmarked operculated cells (that had been sealed overnight) and kept the combs in an incubator so as the larvae could spontaneously emerge from the cells. Adopting this method (described by Nazzi and Milani, 1994) we made sure that the larvae used for artificial infestation were all of the same age (i.e. fifth instar). Moreover, we could exclude any previous infestation, since the mite invades the brood cell just before cell sealing (0-15 hour in the case of worker cells) and spends the first hours after sealing, trapped in the larval food at the bottom of the cells, not interacting with the larva. Please note that according to Donzé and Guerin (“Behavioral attributes and parental care of Varroa mites parasitizing honeybee brood, https://doi.org/10.1007/BF00197001) the mite starts feeding on the bee larva not earlier than six hours after cell sealing. 

To make this point clearer, we reformulated the sentence “L5 honey bee larvae were artificially infested with one mite or maintained uninfested as controls according to Nazzi and Milani”(Lines 492-493 of the original document), as follows: “L5 honey bee larvae were obtained from brood cells capped in the preceding 15 h, as described by Nazzi and Milani [61]; bee larvae were then artificially infested with one mite or maintained uninfested as controls and put into 6.5 mm i.d. gelatin capsules (Agar Scientific Ltd, UK). Since the feeding activity of the mites only starts six hours after cell sealing [62], by using this method, we could exclude that bees classified as uninfested had been infested prior to the experiment. Lines 509-513 of the revised document.

As regards the adult bees used in the first experiment reported in section 4.3, they were collected from sealed brood combs as described in lines 501-505 of the original document. All the bees emerged under lab conditions and were carefully inspected to exclude the presence of possible infesting mites. About 75 individuals were artificially infested with one mite while an equal number was kept mite-free. Of course, we cannot be 100 % sure that bees were not infested during the pupal stage; however, at eclosion, the mature mites tend to stay on the body of the emerging bees (such that dislodging them from there is normally pretty hard); therefore, if they had been infested we should have noticed. Furthermore, when most experiments were carried out, mite infestation in the experimental colonies was only moderate and the proportion of infested brood cells was still low so that a limited number of bees could be naturally infested.

In the second experiment involving adult bees, described in the line 519-522 of the original document, bees were collected as 5thinstar larvae, before a possible mite parasitization (see above). Part of the L5 larvae was artificially infested with one mite and part maintained uninfested as control. All the bees were maintained closed into gelatine capsule under controlled lab conditions throughout the entire developmental period. After 12 days, at the emergence, half of the uninfested bees was infested with one mite while the other half was maintained uninfested as control. 

To make sure that this point is sufficiently clear also for the readers we added some words of clarification in the manuscript. Lines 527-532 of the revised document.

R1.3: Some of the immune genes are down regulated in infested bees - could that be a case of immunosuppression that is often discussed in the Varroa-DWV symbiotic relationship? I am aware that the study was not designed with that intent and that the data presented might not be enough to support strong claims on the matter, but I think it is a valuable element of discussion that could be worth including. However, I will leave that decision to the authors, and will accept if they disregard my comment.

Thank you for this interesting observation.

The consequence of an immunosuppression caused by the Varroa – DWV symbiotic relationship it is actually the most plausible reason for the observed reduced expression of AMPs in the mite infested bees. However, we did not mention this in the discussion because to verify this hypothesis we should have monitored the viral increase in the hours following the infestation in order to demonstrate how, as the viral load increases, the expression of the AMPs increases too and then decrease after surpassing a certain threshold of viral copies.

R1.4: Related to the gene expression analysis – is it correct to compare gene expression between pupae and adults using only one reference gene? Has the reference gene been validated to be constantly expressed between the pupae and imago? Please provide a reference, or some form of data that indicates actin as a suitable reference gene. If no data or reference is available, please add a disclaimer so readers are aware of this problem.

Thank you for this relevant observation.

The choice of actin as the only reference gene in this study derives from the fact that, after testing on pre-imaginal and adult bees, it appeared to be consistently expressed in the dataset with a low standard deviation, a coefficient of variation lower than 5 % and a MFC, the ratio of the maximum and minimum values observed within the dataset, lower than 2 as suggested by de Jonge et al. 

We now cite another reference and added the following sentence: “Relative quantification of genes and DWV was performed adopting with the 2−ΔΔCt method [71] using actin as the housekeeping gene after having verifying its stability in the pre-imaginal and adult bees stages according to de Jonge et al. [72] (coefficient of variation below 5% and ratio of the maximum and minimum values observed within the dataset < 2)”. Lines 658-661 of the revised document.

R1.5: The paragraph lines 506-511 is unclear and needs rephrasing: I understand that the difference between infestation by 1 vs 2 mites is negligible, but I don’t see how it justifies the effect of “accidental fall of the mites”. Please give more rationale.

Thank you for this observation which allows us to better explain the rationale of our choice. 

When we set up the experiment aimed at testing the effect of Varroa infestation at the adult stage, we prepared cages with honey bees individually infested with one mite and observed that occasionally mites can move from one bee to another and also that other mites occasionally got detached from the bees’ body and fall on the cage’s bottom. As a result, for some limited periods of time, bees initially infested with one mite can harbor two mites or even none. Since the possibility of loosing the infesting mites is clearly smaller if bees are infested with two mites and the possibility of becoming uninfested is higher in bees infested with one mite, we tested how this possible circumstance could affect our results through an experiment involving two cages: one containing bees infested with one mite and one containing bees infested with two mites. Since we observed no differences in terms of bees’ survival we inferred that one or two mites (which may sometimes infest the same bee in a cage) do not change the overall effects on the survival of the tested group of bees. 

We added the following sentence in the manuscript: “In principle, adult bees emerging from brood cells could be infested at the pupal stage, however, given the infestation rate of brood cells in the hives used for the experiment when the trial was carried out (about 20 mites/1000 bees), this was likely true only for a small minority of bees. In any case, all the bees used in the experiment were carefully inspected before the artificial infestation and, at eclosion, the mite tends to stay on the body of the emerging bees (often hiding in the petiole area) [63] and it should be noted through a careful visual inspection (lines 527-532 of the revised document) and rephrased the following sentence: “To make sure that the accidental fall of the mites from the infested adult bees did not affect the results of the experiment, we performed a preliminary trial in which adults were infested with one or two mites. Since no differences were observed, in terms of survival, between adult bees infested with one or two mites (Wilcoxon test =0.477, df=1, p=0.49) we considered the possible fall of infesting mites as a seldom un-influent possibility with respect to the assessment of the effects of mite infestation on bee survival.” (lines 506-514 of the original document) as follow: “The accidental detachment of the mites from the infested adult bees and their movement from one bee to another, can result in a little portion of bees being uninfested for some time and others suffering the effects of a double infestation. To make sure that this would not affect our results we carried out a preliminary trial in which adult bees were infested with one or two mites. Since no differences were observed, in terms of survival, between the honey bees of the two groups (Wilcoxon test = 0.477, df = 1, p = 0.49) we considered the possible fall of infesting mites and the occasional double infestation due to mites’ movement as a seldom un-influent possibility with respect to the assessment of the effects of mite infestation on bee survival.” Lines 533-540 of the revised document.

R1.6: Throughout the paper, sample size should be given more clearly for each experiment/plot. I'd suggest adding points to the barplots to get a better idea of data distribution (using function geom_points() in ggplot2 for instance), but the authors can choose to not do that and instead at least adding sample size in relevant plot captions. This is partly why I've ticked "no" in the review form question "Have the authors made all data underlying the findings in their manuscript fully available?". It is also essential that you deposit your raw data on a public repository (e.g. NCBI's SRA). You should also review how you report statistical tests, i.e. italics for things like "df", "p", spaces around "=" signs, etc

Thank you for the useful suggestions. 

We reported the sample size in the plot caption and in every paragraph describing the experiments and we prepared a data file to be attached to the final revised manuscript.

We reviewed the manuscript modifying the spaces around the symbols and adopting the italics where necessary. 

R1.7: line 226-227 of the original document. This statement needs more details, e.g. are you able to plot the distribution of viral loads in Varroa for the samples you characterised? At least give median, mean and standard deviation to be more descriptive.

Thank you for the observation. We added a box and whisker chart in the supplementary materials (S1 fig). 

R1.8: line 287 of the original document. This statement needs a reference. I'm not sure how the artificial feeding of DWV relates to natural inoculation via mite parasitism in terms of quantity of virus transmitted to the bees.

In lines 555-558 of the original document, we justified our choice of administering to the bees a viral amount corresponding to ten to three viral copies by saying that this represents an inoculum as close as possible to that provided by the mite in terms of viral copies.

The references that justify this choice are number 61 and 62. There the authors calculated a contribution to the infection provided by a mite harbouring 10^8 DWV genome equivalents, approximately to about 2500 viral copies (representing the OID50, overt infection dosage). Since the mites collected in our apiary harboured on average 1.44E+08 viral copies (but with a very high variability among the individual mites) we decided to administer to the bees used for the experiment 10^3 DWV viral copies.

We have now slightly modified the text to make this point clearer. Lines 586-592 of the revised document.

R1.9: line375 of the original document. Replace "nice" with something less colloquial, e.g. "well suited" 

Thank you for the suggestion. We replaced “nice” with “well suited”. Line 390 of the revised document.

R1.10: Line 405 of the original document. What is meant by "regardless of the clear response to the parasitic challenge"? Please be more specific.

Thank you for the observation. Actually, the sentence doesn't make sense, we preferred to remove it. 

R1.11: line 428 of the original document. Replace "isn't" by "is not".

Correction done. Line 441 of the revised document.

R1.12: line 561-563. This statement appears too vague to be relevant (and is not referenced). What do the authors mean by "the contribution likely offered by the mite"? I'd think that natural viral inoculation by Varroa is variable and highly dependant on the mites viral load, sure, but also duration of feeding on the bee, etc.

Thank you for the relevant observation. 

As reported above (see R1.8), we administered to bees 10^3 viral copies since we aimed at inoculating a viral dosage similar to that likely transferred by Varroa according to available literature (Gisder et al., 0.1099/vir.0.005579-0 and Möckel et al., 0.1099/vir.0.025940-0) and, moreover, to the infection levels found in our mites. In addition, we tried to provide to the bees a dosage of DWV compatible with what could be acquired via food according to the literature (Schittny et al., 3390/vetsci7030096).

Anyway, we agree that the sentence is too vague; therefore, we rephrased the sentence “In addition, detected amounts of DWV in the commercial honey and pollen are relatively low, between 10^2 and 10^5 [29] while there are no accurate data regarding DWV copy number in the royal jelly although the virus has been identified in the secretions of the hypopharyngeal glands and in the larval food [28,63]. In order to compare the effects of Varroa and contaminated food, we administered to bees a diet containing a viral copy number as close as possible to the contribution to the infection likely offered by the mite (10^3).” (Line 558-563 of the original document) as follows: “Furthermore, the viral load in commercial honey and pollen ranges from 10^2 to 10^6 per g [29,68] while there are no accurate data regarding DWV copy number in the royal jelly, although the virus has been identified in the secretions of the hypopharyngeal glands and in the larval food [28,69]. Therefore, in order to compare the effects of Varroa infestation and the feeding upon contaminated food on viral infection, we administered to each bee 10^3 viral copies by food. “Lines 587-592 of the revised document.

R1.13: line 588 of the original document. In section 4.8 line 588 (and anywhere else relevant), please give the inoculum volume given to larvae in addition to viral titre. is it 5 uL as for the adult bees?

We added the missing information in the material and methods. Line 598 of the revised document.

Reviewer 2

This is a well-written article about the importance of Varroa infestation and DWV infection at different life stages of honey bee. The experimental design was simple but adequate. The results were clear and easy to understand. The discussion was supported by the results and integrated with previous findings. Overall, a high quality manuscript. However, there are a few minor comments that should be easily addressed before publication.

We thank reviewer 2 for her/his positive assessment of our work.

R2.1: Line 167 of the original document: How was Varroa infestation during the pupal stage controlled for in experiment on adult infestation? I'm sure it was part of the protocol but I couldn't find anything about how Varroa were managed at the colony level to make sure there was no impact of pupal infestation on results of the adult bee infestation with Varroa.

In the first experiment with the adult bees, we collected several brood combs containing bees ready to the emergence as described in the line 501-505 of the original document. All the bees emerging under lab conditions were carefully checked for Varroa absence and about 75 individuals were artificially infested with one mite while 75 were kept mite-free.

As the reviewer correctly points out, we cannot be 100% sure that the bees we used were mite free during the pupal stage. However, given the infestation rate of brood cells in the hives used for the experiment when the trial was carried out (about 20 mites/1000 bees), this was likely true only for a small minority of bees. In any case, all the bees used in the experiment were carefully inspected before the artificial infestation and, at eclosion, the mite sits on the body of the emerging bees often in the petiole area. To make this point clearer we have reworded the original sentence as follows:

“In principle, adult bees emerging from brood cells could be infested at the pupal stage, however, given the infestation rate of brood cells in the hives used for the experiment when the trial was carried out (about 20 mites/1000 bees), this was likely true only for a small minority of bees. In any case, all the bees used in the experiment were carefully inspected before the artificial infestation and, at eclosion, the mite tends to stay on the body of the emerging bees (often hiding in the petiole area) [63] and it should be noted through a careful visual inspection”. Lines 527-532 of the revised document.

In the second experiment involving adult bees, described in the line 519-522 of the original document, bees were collected as 5thinstar larvae, before mite parasitization occurs. Part of the L5 larvae was artificially infested with one mite and part maintained uninfested as control. All the bees were maintained into gelatine capsule under controlled lab conditions throughout the entire development period. After 12 days, at the emergence, half of the uninfested bees was infested with one mite while the other half was maintained uninfested as control. 

This second approach was adopted to compare the gene expression level between mite infested pupae and adults.

R2.2: Line 199 of the original document: If the difference is not significant, then this sentence should read "There was no significant difference in DWV level in bees infested with Varroa and the control" or something to that effect.

Thank you for the suggestion. We rephrased the sentence as follow: “In the case of mite infestation at the adult stage, there was no significant difference in DWV level in bees infested with Varroa and the control (Mann-Whitney U test. U = 27, df = 1, P = 0.3) likely because of the high variability observed in the mite-infested group”. Lines 203-205 of the revised document.

R2.3: Line 460 of the original document: What time of year was this study performed? This is important as Varroa populations and viral dynamics in source colonies can significantly influence the outcomes of the experiments.

Thank you for the relevant observation. We performed all the experiments at the turn of spring and summer (from the middle of June to the middle of July) when the mite infestation and the viral levels in our apiary located in the Northeast of Italy are still low but there are enough Varroa mites to conduct the experiments.

We have now added the following sentence to the manuscript to specify the time of the year when the study was performed: “The experiments were carried out from the middle of June to the middle of July, when mite infestation and viral levels in the Northeast of Italy are still low (in the order of 20 mites/1000 bees) but there are enough Varroa mites to conduct the experiments.” Lines 633-635 of the revised document.

R2.4: Line 502 of the original document: This is a call back to the comment on line 167. How was Varroa infestation in the pupal stage controlled for in the experiment on infesting adult bees with Varroa?

See R2.1

R2.5: All the figures were simple, but very effective at showing the data. However, adding indicators of significant differences would make the data much easier to understand visually without having to read the figure legend. This would improve the figures that have multiple treatment groups such as Figures 5, 6, 10, and 11.

Thank you for the suggestion, we added the indicators in the figures.

---

## [Editor Report · Decision Letter 1]

5 Jul 2023

Age-related response to mite parasitization and viral infection in the honey bee 1 suggests a trade-off between growth and immunity

PONE-D-23-04947R1

Dear Dr. Zanni,

I am very happy with the manuscript revisions and your thorough work. Thus, we’re pleased to inform you that your manuscript has been judged scientifically suitable for publication and will be formally accepted for publication once it meets all outstanding technical requirements.

Kind regards,

Olav Rueppell

Academic Editor

PLOS ONE
---

## [Editor Report · Acceptance letter]

7 Jul 2023

PONE-D-23-04947R1 

Age-related response to mite parasitization and viral infection in the honey bee suggests a trade-off between growth and immunity 

Dear Dr. Zanni:

I'm pleased to inform you that your manuscript has been deemed suitable for publication in PLOS ONE. Congratulations! Your manuscript is now with our production department. 

Kind regards, 

on behalf of

Dr. Olav Rueppell 

Academic Editor

PLOS ONE